# Study on the Adsorption Properties of Graphene Oxide/Laponite RD/Chitosan Composites

**DOI:** 10.3390/ma14123224

**Published:** 2021-06-11

**Authors:** Wenjie Du, Rui Ma, Zhiyan Liu, Gang Yang, Tao Chen

**Affiliations:** 1Faculty of Materials Science and Chemistry, China University of Geosciences, Wuhan 430074, China; wenjiedu@cug.edu.cn (W.D.); zhyliu@cug.edu.cn (Z.L.); yg0914@cug.edu.cn (G.Y.); 2Hubei Key Laboratory of Radiation Chemistry and Functional Materials, School of Nuclear Technology and Chemistry & Biology, Hubei University of Science and Technology, Xianning 437100, China

**Keywords:** Graphene oxide/Laponite RD/Chitosan, composites adsorption CO_2_

## Abstract

A novel Graphene oxide/Laponite RD/Chitosan ternary composite was synthesized by sol-gel method and freeze-drying method. The Laponite RD was silanized by 3-aminopropyltriethoxysilane (APTES). Graphene oxide (GO) was prepared by an improved Hummers method. Under the acidic conditions, self-assembly recombination was realized by electrostatic interaction between modified Laponite RD and GO. The results from Fourier transform infrared spectroscopy, X-ray diffraction, and scanning electron microscopy confirmed that the modified Laponite RD was successfully compounded with GO, and the composite is laminated and stacked. The results from BET (Brunauer–Emmett–Teller) methods found that the BET-specific surface area of the hybrid aerogel significantly increased with the increase of the doping content of the composite, and the specific surface area of the aerogel composite with 20% doping content reached 81 m^2^/g. The structure of aerogel is porous, and there are numerous holes in the interior, which is closely related to adsorption properties. Thermogravimetric analysis (TG) test was used to explore the change of thermal properties of hybrid aerogel materials, and it was found that the addition of composite increased the initial decomposition temperature and thermal stability of hybrid aerogel. Finally, the potential applications of aerogel were tested, such as methylene blue adsorption and CO_2_ adsorption.

## 1. Introduction

The discharge of flue gas and wastewater has become a global concern, because many of the colored dyes in the wastewater are not biodegradable, and excessive emissions of flue air such as carbon dioxide will lead to serious climate change [1,2,3,4]. Many localities across China are grappling with extensive groundwater contamination by persistent pollutants. Therefore, the need for the removal of these toxic dyes is urgent from the health point of view. To solve the problem, several strategies have been developed. So far, the methods that can effectively selectively capture CO_2_ from the flue gas mainly include membrane separation and filtration technology and liquid amine washing technology. As for the colorful dyes, the coping strategies are chemical oxidation, membrane filtration, ion exchange and adsorption [5,6,7]. However, most of these strategies are costly and unsustainable. Among them, adsorption has potential application prospect due to its high efficiency, economic feasibility and convenient operation. Aerogel is a kind of nanomaterial with special structure that has many special physical and chemical properties and has potential application prospect in many aspects such as catalyst loading, medical biomaterials, selective adsorption materials and filtration materials [8,9,10]. Aerogel materials have great prospect in selective adsorption because of their low density, high porosity and high specific surface area, which can be used as adsorbent matrix and then loaded with specific groups [11].

He’s team synthesized a solid amine adsorbent prepared by molecular imprinting, and the adsorption performance of CO_2_ has been explored. CO_2_ pre-adsorbed on PEI (polyethyleneimine) could occupy the reactive sites of amino groups and act as a template for imprinting in the cross-linking process [2]. The imino groups formed from the cross-linking reaction between glutaraldehyde and PEI could be reduced by NaBH_4_ to form CO_2_-adsorbable amino groups. The adsorption results indicated that CO_2_ imprinting and reduction of imino groups by NaBH_4_ endowed the adsorbent with a higher CO_2_ adsorption capacity.

Harris and McNeil synthesized one localized hydrogel based on cellulose nanofibers and wood pulp [1]. They developed a rapid, locally formed hydrogel that adsorbs dye during gelation. These hydrogels are derived from cellulose—a renewable, nontoxic, and biodegradable resource. Methylene blue can be adsorbed in large quantities within a few seconds; the maximum adsorption amount can be up to 340 ± 40 mg methylene blue cellulose.

In our laboratory, a class of novel Graphene oxide/Laponite RD/Chitosan ternary composite had been synthesized by sol-gel method and freeze-drying method. There are a lot of oxygen-containing active groups on the surface of GO such as epoxy carboxyl and hydroxyl [12,13]. Moreover, the existence of surface polar functional groups enables them to have large specific surface area and high particle exchange capacity and can react with many functional molecules and chemical groups with specific chemical and biological properties to prepare nanopolymer composite materials, thus effectively improving the comprehensive properties of materials such as thermal, electrical and mechanical properties [14,15]. Chitosan (CT) is a kind of important environmentally friendly natural polymer. It is widely used in medicine, food, water treatment and adsorption. A nanoclay similar to Kaolin, Laponite RD is one of natural inorganic material. Because of its strong absorbability, ion-exchangeability and expansibility, it has good adsorption performance for various types of pollutants in water, so it has broad application prospect in environmental control [16].

In the present work, breaking away from the limitation of conventional CO_2_ adsorbent, a new ternary composite aerogel was synthesized to be used as adsorbent (Scheme 1). Modified Laponite RD and GO were synthesized with Chitosan by electrostatic self-assembly. Aerogel-type solid amine adsorbent has both physical adsorption and chemical adsorption and has excellent adsorption performance [17,18,19]. The adsorption capacity of aerogel was researched. TG test and SEM were carried out to evaluate their applications in the adsorption field. This work will help explore the adsorption mechanism of Chitosan-based aerogels.

## 2. Experimental Section

### 2.1. Materials

Laponite RD was purchased from Rockwood Lithium. 3-aminopropyl triethoxysilane (APTES), hydrochloric acid (37 wt%), methylbenzene, acetic acid and ethanol were purchased from Sinopharm Chemical Reagent Co., Ltd. (Shanghai, China). Commercial grade chitosan was purchased from Geao Chemical Technologies Co., Ltd. (Wuhan, China). The degree of deacetylation of CT was 95%. Glutaraldehyde was purchased from Fuchen Chemical Reagent Factory (Tianjin, China). Graphene oxide (GO) was prepared in the laboratory using an improved Hummers method (which can be found in Appendix A). Other chemical reagents from commercial sources in China were of analytical grade and used without further purifications.

### 2.2. Preparation of the Modified Laponite RD

Laponite RD powders were dispersed into the methylbenzene with stirring for 5 min and sonicating for 30 min. Then, we added 2 mL APTES to the solution with oil bath 110 °C for 24 h. The reactants were separated in a centrifuge at high speed to obtain the organically modified Laponite RD, which was washed three times with methylbenzene and ethanol successively to remove the excess APTES. The product was dried in an electric vacuum drying oven at 40 °C to obtain APTES-modified Laponite RD and was named AP–RD.

### 2.3. Preparation of the GO/Laponite RD/Chitosan Composite Material

The 500 mL 1 g/L AP–RD aqueous solution and 500 mL 1 g/L GO aqueous solution were successively configured. Then, the AP–RD aqueous solution was slowly poured into GO aqueous solution and sonicated for 10 min. The mixed solution was stirred mechanically for 1 h at a rotational speed of 500 r/min; then, 5% diluted hydrochloric acid was added to the solution, the pH of the solution was adjusted to about 2 and the solution was left to stand. After 12 h of settlement, the solution was centrifuged at a speed of 8000 r/min to obtain the precipitation. A large amount of deionized water and multiple centrifugations were used to wash the precipitation to neutral. Finally, the resulting solution was concentrated and quenched in liquid nitrogen, then dried in the freeze dryer for 3 days to obtain tan AP–RD/GO compound, which was ground for use (Figure 1).

When the pH value of aqueous solution is 2, the AP–RD changes from negative to positive due to the interaction between proton and -NH_2_, and Zeta potential is 11.9 mV. The surface of GO was electronegative due to the presence of carboxyl groups, and Zeta potential was −20.9 mV (Table 1).

We took a certain amount of solid AP–RD/GO composites and 1.00 g CT dissolved in 70 mL deionized water and added 1% volume of acetic acid; then, the solution underwent 300~1000 r/min fully stirring and sonicating for 10 min. An appropriate amount of glutaraldehyde (150 μL glutaraldehyde diluted to 1 mL) was added and stirred evenly. After standing for 2 days, it was frozen in liquid nitrogen and dried in a freeze dryer for 3~4 days to obtain GO/AP–RD/Chitosan composite materials with 5–40% doping amount, respectively.

### 2.4. Characterization

Nanometer particle size potential analyzer (Nano ZS90) was used to explore the particle size and Zeta potential. The GO, AP–RD and Laponite RD were dispersed in an aqueous solution for nanometer size analysis and Zeta potential analysis. The volume was 10 mL. The precision for nanometer size analysis is ±2%, and the precision for Zeta potential analysis is 0.12 μm·cm/V·s. To understand the AP–RD/GO/Chitosan composite behavior, Fourier transform infrared spectroscopy (FT-IR) was carried out on Nicolet AVATAR360 with KBr tablet method, and the wave number was in the range of 4000–400 cm^−1^. In order to further clarify the fibrous aggregates of chitosan in the solution, visualized nanofibers were observed by using scanning electron microscopy (SEM). SEM observations of the inner structure of chitosan aerogel were made on a Hitachi S-4000 microscope. XRD measurement was carried out on an XRD diffractometer (AXS-D8-Focus). The XRD patterns with Cu Kα radiation (λ = 0.15406 nm) at 40 kV and 40 mA were recorded in the region of 2θ from 5° to 45°. The samples were ground into powder and dried in a vacuum oven at 60 °C for 48 h. The specific surface area analyzer is an instrument to calculate the specific surface area and pore diameter by measuring the adsorption capacity of the sample to the gas in a fixed specification sample tube. The BET-specific surface area and BJH median pore width were calculated by the standard Brunauer–Emmett–Teller (BET) and Barrett–Joyner–Halenda (BJH) methods. To investigate the adsorption capacity of the samples, specific surface area and pore size analysis were performed on a specific surface area analyzer (V-Sorb 2800P, Beijing). The thermal stability of the sample was analyzed using a comprehensive thermal analyzer (STA 409 PC). In nitrogen atmosphere, the temperature rises from 30 °C to 500 °C at a rate of 10 °C/min. Zeta potential of GO and AP–RD was determined by a Zeta potentiometer, and the compound mechanism was explored.

### 2.5. In Methylene Blue and CO_2_ Adsorption Studies

Using V-SORB 2800P high performance automatic specific surface area and pore size tester, the modified hybrid aerogel (AP–RD/GO–CT (20%)), which has the maximum BET-specific surface area, was subjected to adsorption experiments under the experimental conditions of 298 k and CO_2_ with He.

According to Lambert–Beer’s law, at low concentrations, absorbance is proportional to the concentration of the absorbent. Therefore, the absorbance of the adsorbed solution can be measured after dilution, and then the concentration of the adsorbed solution can be calculated according to the equation of the standard curve, so as to calculate the adsorption amount and removal rate. In this paper, methylene blue solutions with concentrations of 1, 2, 3, 4, 6, 7, 8 × 10^−6^ g/mL were prepared. The absorbance was measured by UV-vis spectrophotometer, and the standard curve was drawn. The equation is:A = 0.1552C R^2^ = 0.997(1)
where R^2^ stand for the degree of fitting. It indicates that the degree of fitting is high. Adsorption experiments for methylene blue were carried out at 298 k. The initial concentration of the methylene blue solution is 100 mg/L, and the dosage of adsorbent was 50 mg. The reaction took 6 h to reach equilibrium.

## 3. Results and Discussion

### 3.1. Study on Modified Laponite RD and GO

Conventional Laponite RD is hard to combine with GO. However, the surface of the modified Laponite RD is positively charged [20,21,22]. The mechanism of modification has been given in Figure 2. The average particle size of Laponite RD is 236.7 nm (Table 1), and the surface is weakly positive. After being modified by the silane coupling agent, the particle size is increased to 264.6 nm due to the grafting of organic groups on the surface, and the particle dispersion index (PDI) of the particles is slightly increased. When pH = 2, the surface of modified Laponite RD is positively charged at 11.9 mV, while the surface of GO is negatively charged due to the presence of oxygen-containing groups such as carboxyl groups. Therefore, when the pH value of the solution is adjusted to 2, the positive property on the surface of modified Laponite RD and the negative property on the surface of GO can be effectively utilized, and electrostatic self-assembly can be effectively realized through the interaction of heterogeneous charges, so as to realize the successful combination of AP–RD and GO. In addition, the hydroxyl group on the surface of RD and the carboxyl group, epoxy group and phenolic hydroxyl group abundant on the surface of GO can also be related through van der Waals force and hydrogen bond [23,24,25].

We silanized Laponite RD with APTES and prepared GO from natural squamous graphite, preparing for the later synthesis of the composite of them. The results from the XRD indicated that alkylation modification did not change the structural characteristics of Laponite RD. -NH_2_ on APTES was successfully grafted to the surface of Laponite RD, and Laponite RD was successfully modified by covalent bond (Figure 2). After organic modification, the particle size of Laponite RD increased slightly, and the surface showed obvious positive electrical properties (Table 1).

Wide-angle X-ray diffraction was carried out to prove the existence of AP–RD and GO in composites. By comparing curves (a) and (b) in the Figure 3, Laponite RD was modified by APTES before and after maintaining crystal structure, and obvious diffraction peaks appeared at 12.6°, 19.8° and 23.7°, corresponding to (001), (020) and (002) crystal face, respectively. The inter-reticular distance is 0.7020 nm, 0.4483 nm and 0.3751 nm, respectively. The difference lies in that the diffraction peak of Laponite RD modified by APTES is somewhat weakened; this may be due to the modification of the APTES, which reduces the Laponite RD molecule order degree. The (001) crystal plane (2 theta = 10.9°) d = 0.8116 nm in curve (c) is the characteristic diffraction peak of GO, which also appears in curve (d). The 10.9° diffraction peak in curve (d) proves the existence of GO in the AP–RD–GO composite material, and the diffraction peak at 12.6°, 19.8° and 23.7° proves the existence of Laponite RD in the AP–RD/GO composite material, indicating that the modified Laponite RD was successfully synthesized with GO, and some crystal structures of both were maintained in the composite material.

Figure 4 indicates that the absorption peak at 3693 cm^−1^ and 3620 cm^−1^ was derived from the stretching vibration peak of Al-OH in RD, while the absorption peak at 910 cm^−1^ was attributed to the bending vibration peak of Al-OH [26,27]. In contrast to curve (a), curve (b) shows APTES modification in the infrared spectra and a generally consistent curve. The main difference is that 1470 cm^−1^ and 2935 cm^−1^ of the characteristic absorption peaks respectively belonged to methylene -CH_2_ stretching vibration and bending vibration peak, and 1566 cm^−1^ and 3430 cm^−1^ of the absorption peaks belonged to -NH_2_ expansion and bending vibration peak, respectively, [28,29,30] showing that APTES on -NH_2_ successfully grafted to the surface of RD. Laponite RD was successfully modified by covalent bond. In the infrared spectrum of curve (c), a large number of stretching vibration characteristic peaks of -OH appear in 3621–3406 cm^−1^, absorption peaks at 1726 cm^−1^, 1621 cm^−1^ and 1054 cm^−1^ are respectively attributed to the stretching vibration peaks of C=O, C=C and C-O bonds of GO, and the characteristic peak of GO epoxy group is at 1100 cm^−1^ [31,32,33], indicating the successful preparation of GO with rich chemical groups. Curve (d) is the infrared spectrum of RD/GO complex, and the characteristic peaks in both curve (b) and (c) appear in curve (d), indicating that RD and GO were successfully recombined. This is because the silane coupling agent APTES is successfully grafted to the surface of Laponite RD through hydrolysis, so as to effectively modify the surface of Laponite RD with negative charge.

### 3.2. Construction and Structure of the GO/AP–RD/Chitosan Composite

To investigate the BET-specific surface area, BJH adsorption cumulative volume and BJH median pore width of Chitosan aerogels doped with different compounds, different doping contents were set as the control group. Table 2 is the result of the experiment. BET-specific surface area of the aerogel as AP–RD/GO content showed a trend of increase. It reached the highest when the doping amount was 20%. This could be the Chitosan molecular chain on the presence of large amounts of free amino and carboxyl. Those pairs of groups can be combined with AP–RD/GO composites surface hydroxyl and other groups easily through ion exchange and van der Waals force of adsorption on the surface of the inner and outer layers of AP–RD/GO composites, so that the BET-specific surface area of the composite material increased, but, after excessive compounds with Chitosan caused the clogging in structure, BET-specific surface area decreased.

Information related to porous properties of the aerogels was obtained from their respective nitrogen adsorption isotherms recorded at 77 K. The BET-specific surface area, BJH adsorption cumulative volume and BJH median pore width are listed in Table 2. Moreover, the results (Figure 5) from N_2_ and He adsorption–desorption curve of aerogel indicate that it is a typical type IV H3 adsorption desorption isotherm published by IUPAC, and there is a hysteresis loop phenomenon. It indicates that the sample has a typical flap-like grain slot cavity structure, and capillary condensation occurs [34,35]. The results from SEM (Figure 6) also prove aerogel has a loose, porous structure. In the low-pressure section (P/P_0_ = 0–0.01), the isotherm exhibits a sharp adsorption trend, indicating the presence of micropores in the aerogel. After the pressure increases (P/P_0_ = 0.1–0.9), the adsorption capacity increases gradually and is accompanied by a hysteresis loop. At this stage, the change in the adsorption capacity can be used as a basis for measuring the aperture. The adsorption isotherm did not increase significantly under 0.1–0.9 relative pressure, while the adsorption capacity increased sharply under greater than 0.9 relative pressure, indicating that a large number of mesoporous structures existed in the hybrid aerogel [36]. To summarize, the pore structure of aerogels is essentially hierarchical, with pore sizes ranging from microporous to mesoporous. The aperture is between 0 and 50 nm, high porosity, and the combination of compounds with Chitosan is relatively strong, and the complex fully embedded into the Chitosan base surface. The composite is almost invisible on the SEM (Figure 6a,b), and this structure is conducive to dye such as adsorbate in its adsorbent inside, so as to improve the adsorption ability of aerogel.

To determine the stability of the Chitosan gel in different temperature, thermogravimetric analysis was carried out to investigate the weightlessness. Figure 7 shows the effects of the doping amount of the RD/GO. There are mainly two decomposition temperatures in the range of room temperature up to 500 °C. The first weight loss stage occurred in the range of room temperature to 100 °C, mainly due to the evaporation of residual water on the surface or inside of the sample. The second weight loss occurred in the range of 300~400 °C, mainly due to the decomposition of Chitosan and the fracture of aerogel molecular chains. It can be seen in Sample ABCDE curves (F can be seen in Appendix A) that, with the increase of the complex doping content, the second phase of the initial temperature increased; this may be because of the complex formed, and the Chitosan molecular chain enhanced, even interpenetrating the network structure. Moreover, GO, itself, has excellent thermal properties. The synergistic effect of the two improved the thermal stability of hybrid aerogels. The temperature in the flue environment where the adsorbent is used is generally not more than 120 °C. Therefore, this kind of composite material has a certain application prospect.

### 3.3. Adsorption Properties of Aerogels

Methylene blue was used as the adsorption object to test the adsorption property of the sample (AP–RD/GO–CT (20%)), which has the maximum BET-specific surface area. Methylene blue (C_16_H_18_CIN_3_S) is one of the most common cationic organic dyes [37,38]. It is widely used in chemical indicators, dyes, biological dyes and pharmaceuticals. According to Lambert–Beer’s law, at low concentrations, absorbance is proportional to the concentration of the absorbent [39,40,41]. Therefore, the absorbance of the adsorbed solution can be measured after dilution, and then the concentration of the adsorbed solution can be calculated according to the equation of the standard curve, so as to calculate the adsorption amount. The equation is A = 0.1552C (Figure 8).

It can be seen from Figure 9 that the adsorption capacity of the sample (AP–RD/GO–CT (20%)) reached 296 mg/g at t = 5 min. The adsorption capacity increased rapidly with time in the early stage, reached 80% of the maximum adsorption capacity at 50 min, and increased relatively slowly after 70 min, reaching the maximum adsorption capacity of 436.2 mg/g at T = 220 min.

The adsorption process of CO_2_ was carried out at 298 K and from 0 up to 0.1 MPa. The adsorption capacity of hybrid modified aerogel (AP–RD/GO–CT (20%)), which has the maximum BET-specific surface area, was tested (Figure 10). The gravimetric capture of CO_2_ for sample (AP–RD/GO–CT (20%)) is 78.9 mg/g at 298 K and 0.1 MPa. It is generally accepted that CO_2_ capture capacity is determined by numerous factors such as surface area, pore size and pore function, etc., each of which carries different weights at different pressures and temperatures. It is an essential strategy to form a strong interaction between the polymer network and CO_2_. Moreover, if its surface is CO_2_ - philic, such a polymer porous material will exhibit more efficient CO_2_ absorption capacity [42]. The chemical functionalization of porous material with polar groups (such as nitrogen-rich groups, oxygen-rich groups and inorganic ions) can enhance the average dipole–quadruple interactions with CO_2_, thus improving the CO_2_ capture capacity [43]. In this work, the amino group in chitosan and various polar groups in GO can also improve the adsorption capacity of CO_2_. At low pressures in Figure 10, the interaction between CO_2_ and pore surface plays a leading role in CO_2_ capture. The CO_2_ adsorption isotherm of the aerogels rises sharply in low pressure range, indicating that the amino present in aerogels have favorable binding energy with CO_2_ molecules. When the pressure increased to 0.1 MPa, the effect of functionalization to CO_2_ capture gradually weakened. In the meantime, the effect of BET surface area gradually increased. This is also the reason for selecting sample AP–RD/GO–CT (20%).

## 4. Conclusions

A novel ternary composite gel was synthesized. The system composed of the modified Laponite RD, chitosan and GO. The properties of modified Laponite RD and GO complex were investigated. The results from the XRD and FT-IR indicated that Laponite RD was successfully synthesized with GO, and some of their crystal structure was maintained in the composite. The characterization of the chitosan/Laponite RD/GO composite aerogel showed that the specific surface area and pore volume of aerogel demonstrated an increasing trend with the increase of Laponite RD/GO content, reaching the maximum at 20%, and the average pore size remained basically unchanged. SEM photographs were used to observe that the hybrid aerogels showed obvious loose and porous structure with a large number of holes on the surface and in the interior. TG test was used to study the thermal properties of hybrid aerogel materials, and it was found that the addition of composites increased the initial decomposition temperature of hybrid aerogel and enhanced its thermal stability.

The adsorption test of methylene blue and CO_2_ on the modified hybrid aerogel material showed that the initial adsorption capacity of methylene blue reached 436 mg/g at 220 min, and the adsorption capacity of CO_2_ reached 78.9 mg/g. Compared with the available adsorbents, this adsorbent has several advantages such as low cost of raw materials, environmental friendliness and relatively high adsorption capacity. To summarize, we successfully prepared GO/Laponite RD/Chitosan composite materials by sol-gel and freeze-drying methods. Its application in the adsorption of dyes as well as CO_2_ separation and adsorption has a certain practical significance and has a good industrial application prospect.

## Data Availability

Not applicable.

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
