# Peer review of "Study on the Adsorption Properties of Graphene Oxide/Laponite RD/Chitosan Composites"

_materials, 2021, doi:10.3390/ma14123224_

Round 1

Reviewer 1 Report

Journal: Materials

Manuscript ID: materials-1228698

Type: Article

Title: Study On The Adsorption Properties Of Graphene Oxide /Laponite RD/

Chitosan Composites

The authors have described the composite preparation of graphene oxide/Laponite RD/ Chitosan. The composite was characterized well. This composite was used for the adsorption of methylene blue and CO2. The discussion should be improved for better understanding.  Thus, I suggest the manuscript for publication after considering the below points.

  • Abbreviations are used in the abstract, experimental section, and result and discussion without explanations. Examples, TG, AP-RD, BET, etc
  • Abbreviation of the composite should be mentioned.
  • What is the role of APTES?
  • Figures 1, 2, 6, and 7 are mentioned in the text.
  • Figure 5 is missing.
  • Zeta potential of composite is missing.
  • Constant sample names should be included; abbreviations are confusing because different abbreviations are used in different places.
  • In the XRD pattern of RD-GO, GO patterns are small than laponite RD why?
  • In Figure 4, the labels are mentioned as a, b, c, d but in the text curve, A, B, C, D are included. This should be corrected.
  • In Construction and structure of the Go/Laponite RD/Chitosan Composite section-Table 2 contains only CT and RD/GO-CT. The results for the composites where APTES included are missing.
  • Why only the adsorption-desorption isotherm of sample RD/GO-CT(20%) is included.
  • In adsorption properties of Gels, what are Gels?
  • What are the advantages of these composites towards the adsorption of CO2 than compared to available CO2 adsorbent composites?

Author Response

Dear Editor:

We would like to express our gratitude for the editor’s instructive suggestion and comments on our manuscript entitled “Study On The Adsorption Properties Of Graphene Oxide /Laponite RD/ Chitosan Composites” (materials-1228698). According to your elaborative comments, we have revised our manuscript carefully. The revised portion of the manuscript has been indicated in “Track Changes” for quick attention of you. The specific revision process is as follows:

To Referee 1

1and 2. According to your suggestion, the abbreviations are explained when they first appear.

  1. 3. APTES is one of the silane coupling agents, and it hydrolyzes in water. APTES were used to modify the Laponite RD to make it more positively charged, so that it could be combined with negatively charged GO through electrostatic interaction.
  2. After checking, all the figures are referenced correctly.
  3. 5. Figure 5 has been relabeled.
  4. 6. The reason why we did not explore the Zeta potential of the composites is that conventional Laponite RD is hardly combine with GO because of the electrically charged, and Zeta potential did prove that. Moreover, Zeta potential is just a proof of the success of modification. Therefore, the Zeta potential of composite materials was not characterized.
  5. 7. According to your suggestion, the sample names have been corrected.
  6. 8. On the one hand, the content of GO is less than Laponite RD in composites. However, a more likely reason is the combination with Laponite RD, which reduces the degree of order in GO molecules.
  7. 9. According to your suggestion, it has been corrected to “a, b, c, d”.
  8. 1 APTES is just used to modify RD, but not as a component of composite materials.
  9. 1 All the isotherms have been given in page 10. Moreover, the adsorption and desorption isotherms of nitrogen were redetermined carefully.
  10. 1 The sample (AP-RD/GO-CT (20%)) which have the maximum BET specific surface area was selected to explore the adsorption properties. Besides, the selected sample is aerogel.
  11. 1 According to your suggestion, the advantages have been given briefly in line 357.

Reviewer 2 Report

The manuscript by Du et al. reports on the synthesis and characterization of new ternary composites. The overall message is clear and concise, and the experiments are designed well. The language is fine, however, I have several critical comments. All the abbreviations should be defined at first mention, especially in the abstract (BET - line 15, TG - line 19). Besides, there are some typos in the text, e.g. missing spaces (line 45, 88, 89, 131, 156), errors in capital and lowercase letters (e.g. "Go" should be replaced with "GO" in line 87, AL-OH with Al-OH in line 197), some obvious mistakes ("flue air" instead of "flue gas" in line 26 or "saponite" instead of "laponite" in line 73). The grammar should be refined at some places (e.g. "across the China" should be replaced with "across China" in line 29). At parts of the text it was impossible to tell what the authors were trying to convey, e.g. in lines 105-107: "After 12 h of settlement, the settlement was centrifuged at a speed of 8000 r/min to obtain the settlement". Careful spell, typo, and grammar check including clarifying the manuscript where needed is strongly advised.

There are also more specific comments. Table 1 is not referenced anywhere in the text. Please explain in detail how the data reported in Table 1 were determined and supplement the numeric values with their relevant standard uncertainties. Besides, the analysis of XRD patterns (lines 182-193) should be rewritten in a more focused and understandable form. I am attaching a modified Figure 3 with vertical dashed lines that help to follow specific reflections attributed to different investigated phases. Besides, a weak but evident reflection is observed at ca. 20 deg in the diffractograms of unmodified and modified Laponite RD (plots a and b) and the final AP-RD-GO material (plot d). Please comment on that. Finally, the authors should elaborate on the adsorption of CO2 (lines 278-291). This is the most important part of the research, due to its potential applications. More details should be given on the experimental methods (concentration of CO2 in He mixture, flow rate, etc.), in particular in view of the practical use of the synthesized material.

Author Response

To Referee 2:

1.According to your suggestion, all the abbreviations has been defined at first Mention, such as “BET, TG, PDI”.

  1. The errors of missing spaces (line 45, 88, 89, 131, 156), capital and lowercase letters (line 87, 197) has been corrected.
  2. 3. The mistakes in line 26, 73 has been corrected.
  3. 4. According to your suggestion, the revised sentences are as follows “Many localities across China are grappling with extensive groundwater contamination by persistent pollutants” (line 29).
  4. 5. The revised sentences are as follows “After 12 h of settlement, the solution was centrifuged at a speed of 8000 r/min to obtain the precipitation”.
  5. 6. Table 1 has been referenced at line 168 in the revised manuscript.
  6. 7. The numeric value in table 1 was measured by a nanometer particle size potential analyzer (Nano ZS90). The GO, AP-RD and Laponite RD were dispersed in an aqueous solution for nanometer size analysis and Zeta potential analysis. Besides, the volume is 10 ml. And the precision for nanometer size analysis is ±2%, the precision for Zeta potential analysis is 0.12um.cm/V.s.
  7. 1 According to your suggestion, some parts of the XRD analysis has been rewritten.  I really appreciate your modification to figure 3. Besides, the diffraction peaks appears at 19.8° is corresponding to (020) crystal face of Laponite RD. As you said, it is too weak, so it wasn't marked out in the previous manuscript.
  8. 1 According to your suggestion, the parts of the CO2adsorption have been rewritten (line 323-342). More details have been given such as theisotherm of the CO2 adsorption from 0-0.1 MPa. Some factors affecting the adsorption performance of CO2 are also discussed.

Reviewer 3 Report

The paper presents the studies on the adsorption properties of graphene oxide/laponite RD/chitosan composites. The potential applications of composite materials were tested, i.e. methylene blue and/or CO2 adsorption. The methods' presentation and scientific results are satisfactory for publication in the Materials journal. The minor and major drawbacks to be addressed can be specified as follows:
1.    Page 1, line 44. (i) Hui He’s team ---> He’s team. (ii) Reference(s)?
2.    Page 1, line 45. PEI? Please, explain it.
3.    Page 2, line 51. (ii) Justin T. Harris and coworkers ---> Harris and McNeil. (ii) Reference(s)?
4.    Page 2, Scheme 1. Glutaraldehyde is poorly visible in the hydrogel.
5.    The authors have to decide whether the word “laponite RD” is capitalized or lower case. Likewise “chitosan”. AP – RD or AP – Laponite RD. Please standardize it.
6.    Page 4, lines 133 and 134, “pore diameter by measuring the adsorption capacity of the sample to the gas in 133 a fixed specification sample tube.” Analyzing the average pore diameter values (Tab. 2) and the adsorption isotherms (Fig. 6), I must say that these values are strange and absurd. Unfortunately, there are too few details given in the paper to explain why. Please discuss how exactly these values corresponded. Were they obtained from pore distribution? The theoretical description of the adsorption isotherms? Equation?
7.    Page 5, line 156 and all the manuscript. A lot of typos, for example, “Conventionallaponite”.
8.    Page 5, lines 158 and 171, Tab. 1. What is “surface particle size”? The authors used “the average particle size “ in line 158. The second one is better.
9.    Page 5, line 171, “of RD and GO”. What about AP-RD?
10.    Page 5, Tab. 1. (i) Too high accuracy of numbers, i.e. 236.7 ---> 236. (ii) PDI – please explain this shortcut. (iii) Zeta Potential ---> Zeta potential.
11.    Pages and 6, Figs. 3 and 4. The different sample nomenclature, i.e. (AP – RD or AP – Laponite RD) and (AP – RD/GO  or RD-GO). See all the manuscript.
12.    Page 7, Tab. 2. (i) Too high accuracy of numbers for pore volume, i.e.0.2145 ---> 0.214. (ii) average pore diameter.
13.    Did the authors determine the densities? It was worth comparing this quantity with other aerogels.
14.    Page 8, Fig. 6. Please publish all isotherms.
15.    Page 8, Fig. 7. What samples?
16.    Page 9, Fig. 8. Please add results for other samples (see Tab. 2).
17.    Page 10, Fig. 9 and 10. What samples?
18.    In my opinion, the results (adsorption isotherms and kinetics curves) for the selected samples should be collected in the supporting information.

Author Response

To referee 3

  1. According to your suggestion, the revised sentences are as followed “He’s team synthesized a…” and it can be found in reference 2.
  2. The abbreviation “PEI” has been explained in line 47.
  3. According to your suggestion, the revised sentences are as followed “Harris and McNeil synthesized…” and it can be found in reference 1.
  4. 4. According to your suggestion, page 3, scheme 1. The picture has been modified in the revised manuscript. The glutaraldehyde is more visible.
  5. 5. The words have been corrected to “Laponite RD, Chitosan, AP-RD” in the revised manuscript.
  6. 6. I am so sorry to say that there may be some errors such as air leakage or too little sample mass in the process of adsorption desorption experiment. So, we repeated the nitrogen adsorption desorption experiment and recalculated the BET specific surface area, BJH median pore width and BJH adsorption cumulative volume. And the adsorption desorption isotherms were discussed in details at page 10. Besides, the Table 2 has been relabeled.
  7. 7. The errors such as “missing spaces, typos” has been corrected in the revised manuscript.
  8. 8. According to your suggestion, the word “surface particle size” has been corrected to “the average particle size”.
  9. The sentence were rewritten as “Average particle size, PDI and Zeta potential of Laponite RD, AP-RD and GO” in line 185-186.
  10. The accuracy of numbers have been corrected in Table 1. And the shortcut “PDI” has been explained in line 174 and 187-188. The word “Zeta Potential” has been corrected to “Zeta potential” in Table 1.
  11. 1 The sample nomenclature has been unified, such as “AP-RD, AP-RD/GO”.
  12. 1 The accuracy of numbers have been corrected in Table 2.
  13. 1 The true density of the sample (AP-RD/GO-CT (20%)) was measured by the specific surface area analyzer. The more details were given in supporting information.
  14. 1 We repeated the nitrogen adsorption desorption experiment and recalculated the BET specific surface area, BJH median pore width and BJH adsorption cumulative volume. All the isotherm has been published, and it was discussed in detail.
  15. 1 The SEM images of aerogel is AP-RD/GO-CT (20%), which have the maximum BET specific surface area.
  16. 1More experimental results have been added in the form of supporting information. Besides, Table 2 was rewritten in the revised manuscript.
  17. 1 Figure 8 is the standard operating curve of methylene blue solutions, it was measured by set a series of concentrations of methylene blue solution and measured its absorbance. Finally, a straight line is fitted. Fig. 8 and 9 is for the sample AP-RD/GO-CT (20%).
  18. 1 The details of N2adsorption desorption isotherm has been given at page 10 line 251-272 in revised manuscript. The thermogravimetric analysis curve F for sample AP-RD/GO-CT (30%) have been added in the form of supporting information.

Reviewer 4 Report

The authors present the adsorption properties of the prepared graphene oxide/laponite RD/ chitosan composites. The work is interesting but before publication it needs some revision:

  1. English language needs improvement.
  2. Figure numbering is incorrect (after figure 4 is figure 6; Figure 10 is twice numbered).
  3. In some parts of the manuscript the authors mention saponite instead of laponite. This should be corrected.
  4. Abbreviations are inconsistent: for example for chitosan they use both CT and CTS. 
  5. The GO synthesis method is not presented. 
  6. What kind of oil did they use in the adsorption studies (In the Conclusion part they stated: "Its application in the adsorption of dyes and oil...".

Author Response

To referee 4

1.All the errors such as “typo and abbreviation” has been corrected in the revised manuscript.

  1. According to your suggestion, the figure numbering has been corrected.
  2. 3. The word “saponite” has been corrected to “Laponite RD”
  3. 4. The abbreviations such as “CT, Laponite RD” have been unified.
  4. 5. According to your suggestion, the GO synthesis method has been presented in reference 41.
  5. 6. We didn't do any experiments on oil absorption, so, the word “oil” has been deleted.

Round 2

Reviewer 3 Report

The authors have made a substantial improvement for this article. The manuscript can be accepted for publishment in the present form.

Author Response

We have improved our manuscript according to your suggestions.

Reviewer 4 Report

The manuscript was improved.

However I still have some minor observations:

-in Scheme 1 appears "saponite", please correct.

- Add a short description on GO synthesis in supplementary files.

Author Response

Thanks for your suggestion! We have corrected "saponite" in Scheme 1 , and added a short description on GO synthesis in supplementary files.
